# Influenza vaccination coverage among emergency department personnel is associated with perception of vaccination and side effects, vaccination availability on site and the COVID-19 pandemic

**Anna-Maria Stöckeler[1], Philipp Schuster[2], Markus Zimmermann[3], Frank Hanses[1,3]***

1 Department for Infectious Diseases and Infection Control, University Hospital Regensburg, Regensburg, Germany, 2 Institute of Medical Microbiology and Hygiene, University of Regensburg, Regensburg, Germany, 3 Emergency Department, University Hospital Regensburg, Regensburg, Germany

* frank.hanses@ukr.de

## Abstract

### Introduction

Influenza is a major concern in hospitals, including the emergency department (ED), mainly because of a high risk for ED personnel to acquire and transmit the disease. Although influenza vaccination is recommended for health care workers, vaccination coverage is low.

### Methods

This survey was conducted in the 2016/2017 and 2020/2021 influenza seasons. Questionnaires were sent to ED personnel in 12 hospitals in Bavaria, South-Eastern Germany. The response rates were 62% and 38% in 2016/2017 and 2020/2021, respectively. Data were compared between the two seasons as well as between vaccinated and not vaccinated respondents in 2020/2021.

### Results

Significantly more ED personnel reported having been vaccinated in the 2020/2021 season. Factors associated with vaccination coverage (or the intention to get vaccinated) were profession (physician / medical student), having been vaccinated at least twice, the availability of an influenza vaccination on site (in the ED) as well as the COVID-19 pandemic. Additionally, significant differences in the assessment and evaluation of influenza, its vaccination side effects and ethical aspects were found between vaccinated and not vaccinated ED personnel in 2020/2021. Unvaccinated respondents estimated higher frequencies of almost all potential vaccination side effects, were less likely to accept lay-offs if employees would not come to work during an influenza pandemic and more likely to agree that work attendance should be an employee´s decision. Vaccinated participants instead, rather agreed that

**Data Availability Statement:** An anonymized version of the dataset is available at zenodo.org (https://doi.org/10.5281/zenodo.5164388).

**Funding:** The authors received no specific funding for this work.

**Competing interests:** The authors have declared that no competing interests exist.

vaccination should be mandatory and were less likely to consider job changes in case of a mandatory vaccination policy.

## Conclusion

The COVID-19 pandemic might have contributed to a higher influenza vaccination rate among ED workers. Vaccination on site and interventions targeting the perception of influenza vaccination and its side effects may be most promising to increase the vaccination coverage among ED personnel.

## Introduction

Seasonal influenza is an acute contagious respiratory disease caused by type A or B influenza viruses [1]. The disease is associated with significant morbidity causing a high number of cases, hospitalizations and deaths annually [2]. Each year, approximately 4–16 million people contract influenza in Germany, of which up to 20000 die [3]. Vaccination represents a cost-effective way to reduce influenza-like illnesses, sick leave, societal cost [4, 5] and is deemed especially important for persons at high risk of contracting the virus or becoming severely ill [6–8]. Health care workers are frequently exposed to contagious agents leading to a risk of infection and transmission of diseases [9]. In particular, ED staff are at the front line caring for a large number of patients compared with other departments, including both vulnerable patients and individuals with acute respiratory infections like influenza. Not least due to the close proximity and an extended duration of patient contacts, the ED poses a great risk for spreading nosocomial infections [9–11]. Besides vaccination of persons with increased risk of complication from influenza disease, WHO, CDC as well as the German standing vaccination committee (STIKO) recommend that health care workers among others get vaccinated annually against influenza to protect employees, decrease work absenteeism and reduce virus spreading and transmission within the hospital [7, 12–14]. However, vaccination rates among health care workers in general and ED personnel in particular fall short of expectations and implementation strategies have led to mixed success so far. According to a survey by the Robert Koch Institute (the institute for Public Health in Germany–RKI) conducted in 52 hospitals including 5808 employees, only 39,5% of health care workers in Germany were vaccinated in the 2017/2018 season [12, 15]. Previous research into the reasons why health care workers choose not to get vaccinated has highlighted concerns about its effectiveness or lack of knowledge about influenza vaccination [11, 16, 17]. The exposure to severe respiratory infectious diseases during the current COVID-19 pandemic may change the perceptions and positively affect the influenza vaccination rate [16]. We hypothesize that higher uptake of influenza vaccination can be found compared to previous years among health care workers, especially those working in the ED and thus are frequently exposed to COVID-19 patients. We therefore conducted a survey among ED personnel from different hospitals in Bavaria, South-Eastern Germany, on influenza and influenza vaccination and compared influenza vaccination attitude/readiness from a "before COVID-19" season (2016/2017) to a "during COVID-19" season (2020/2021).

## Materials and methods

Our impression was that the vaccination rate in emergency departments leaves room for improvement. We therefore conducted a baseline survey between March 2, 2017 and July 3, 2017 (influenza season 2016/2017) among ED personnel in hospitals in Bavaria, South Eastern

Germany, with a planned later follow-up to monitor changes in attitudes with regard to influenza and influenza vaccination over time. We compared these results in the context of the current SARS-CoV-2 pandemic with a survey conducted between November, 16, 2020 and January 15, 2021 (influenza season 2020/2021). In total, EDs from 14 hospitals were invited to participate in the survey and answers were received from 11 and 12 departments in 2016/2017 and 2020/2021, respectively (Universitätsklinikum Regensburg, Klinikum Memmingen, Klinikum Weiden, München Klinik Harlaching, Klinikum Rosenheim, Klinikum Augsburg, Klinikum St. Marien Amberg, Klinikum Dritter Orden München, Klinikum Kempten, Klinikum St. Elisabeth Straubing, Krankenhaus Cham, Klinikum Ingolstadt). Additional questions were asked in the survey of 2020/2021 regarding the current COVID-19 pandemic and its potential impact on influenza vaccination. A full sample questionnaire (in German) can be obtained from the authors. The study was approved by the ethics committee of the University of Regensburg (IRB number 19-1631-101).

## Sample recruitment and data collection

Sample recruitment and data collection followed the same procedure in both seasons. The sample consisted of emergency department personnel working in Bavarian hospitals (including university hospitals, municipal/country hospitals and hospitals under church sponsorship, with a capacity ranging from 480 to 1100 beds). The average annual number of ED patients ranged from 26000 to 48000. Physicians (and medical students), as well as nurses and others (mostly administrative personnel) participated in the survey. Participation was voluntary and uncompensated, and the questionnaire was anonymous and self-administered on a paper-pencil basis. Initially, the heads of the emergency departments in various hospitals in Bavaria were asked to participate in the study. After positive feedback, version 1 of the questionnaire (including questions on ED size and the requested number of questionnaires for participating personnel) was sent to the ED heads. Based in the information obtained herein, version 2 of the questionnaire (for ED personnel itself) was sent to the leaders of the emergency departments and distributed locally by them. In total, 727 questionnaires were sent to the different hospitals. The response rate was calculated from the total number of completed questionnaires returned back (142 and 190 in 2016/2017 and 2020/2021, respectively) divided by the total number of questionnaires sent out (229 in 2016/2017 and 498 in 2020/2021). Contacted EDs with no completed survey from ED staff were excluded (n = 2).

## Query and variables

The questionnaire was loosely built on previous work on influenza vaccination and perceptions among hospital personnel [18, 19]. It included the categories 'influenza', 'influenza vaccination' and 'socio-demographics'. For the 2020/2021 season, the category 'impact of COVID-19 on influenza vaccination' was added. In the 'influenza' category, the frequency of the disease and its complications as well as the impact on the ED were asked to be assessed on a 5-Point Likert scale. The listed side effects are a selection based on previous studies [19]. The 'influenza vaccination' category comprised knowledge questions on vaccination statements and the current recommendations of the STIKO on a binary scale as well as an estimation of vaccination side effects on a 6-Point Likert scale. Additionally, participants were asked about their influenza vaccination history and reasons for/against the vaccination (single choice), if they would recommend influenza vaccination to colleagues (binary scale), the future handling of the influenza vaccination in EDs (5-Point Likert scale), the current vaccination situation in their EDs (binary scale) and some ethical questions about influenza and vaccination (5-Point Likert scale). The category 'impact of COVID-19 on influenza vaccination' assessed a possible impact

of the COVID-19 pandemic on influenza vaccination rate (binary scale). Socio-demographic questions asked included gender, age, profession and years working in the ED (single choice).

## Statistical analysis

Statistical analysis was carried out using R version 3.6.2. Groups were compared using Mann-Whitney U tests for ordinal or continuous variables and chi-square tests for categorical variables. Logistic regression was used to compute odds ratios in a multivariable model. Survey items on a Likert scale were analyzed using the 'likert' package in R [20]. Differences in the answer distributions between two groups were explored by comparing means using Mann-Whitney U tests after converting ordinal answer levels to a numeric scale [21]. P-values <0.05 were considered significant. The comparison between vaccinated and unvaccinated participants was limited to the 2020/2021 season to ensure that all variables were available for analysis including the items pertaining to the SARS-CoV-2 pandemic.

## Results

In total, 332 ED personnel completed the survey, 142 (43%) of these in the 2016/2017 season and 190 (57%) in 2020/2021. Response rates were 62% and 38% in 2016/2017 and 2020/2021, respectively (ranging from 23%–100% in 2016/2017 and from 16%–72% in 2020/2021 per ED). Locations of participating EDs are depicted in Fig 1.

In both years, more surveys were completed by women and nurses. There were no statistically significant differences, however, among respondents in both seasons regarding gender, older age (> 44 years) and profession (Table 1).

Of the 332 respondents overall, 156 (47%) reported having been vaccinated against influenza in the respective season, while 176 (53%) had no current vaccination. Comparing the two seasons, both the vaccination rate in the respective season as well as the rate of ED personnel that reported at least two vaccinations were significantly higher in 2020/2021. Also, significantly more respondents reported that vaccinations were offered on site (in the ED) in 2020/2021.

In order to explore the observed increase in influenza vaccination coverage, factors with a significance level of p<0.1 in the comparison between both seasons were carried over to compare vaccinated and unvaccinated respondents in the 2020/2021 season. Vaccination offered on site, being vaccinated at least twice and profession (physician or medical student) as well as the intention to get vaccinated because of COVID-19 were associated with a significantly higher vaccination coverage in 2020/2021 in univariate and multivariate analysis (Table 1).

Most questionnaire items related to influenza incidence and complications revealed no significant differences between 2016/2017 and 2020/2021 seasons as well as between vaccinated and not vaccinated respondents in 2020/2021. The only exceptions were that, compared to 2016/2017, 2020/2021 respondents estimated a higher influenza death rate whereas the infection rate among nurses was estimated to be higher by vaccinated respondents than by unvaccinated participants in 2020/2021 (Fig 2).

The perception of side effects associated with influenza vaccination did not change between the two seasons. In contrast, unvaccinated personnel (2020/2021) consistently estimated significantly higher frequencies of all potential vaccination side effects listed in the survey (headache, body aches, fever, shivering, skin necrosis, unable to work, encephalitis)—with the notable exception of local pain at the injection site that was estimated to occur frequently by both groups (Fig 3).

With regard to the burden of influenza on the ED, more participants from 2020/2021 agreed that influenza poses a serious threat for ED patients. Vaccinated participants in 2020/

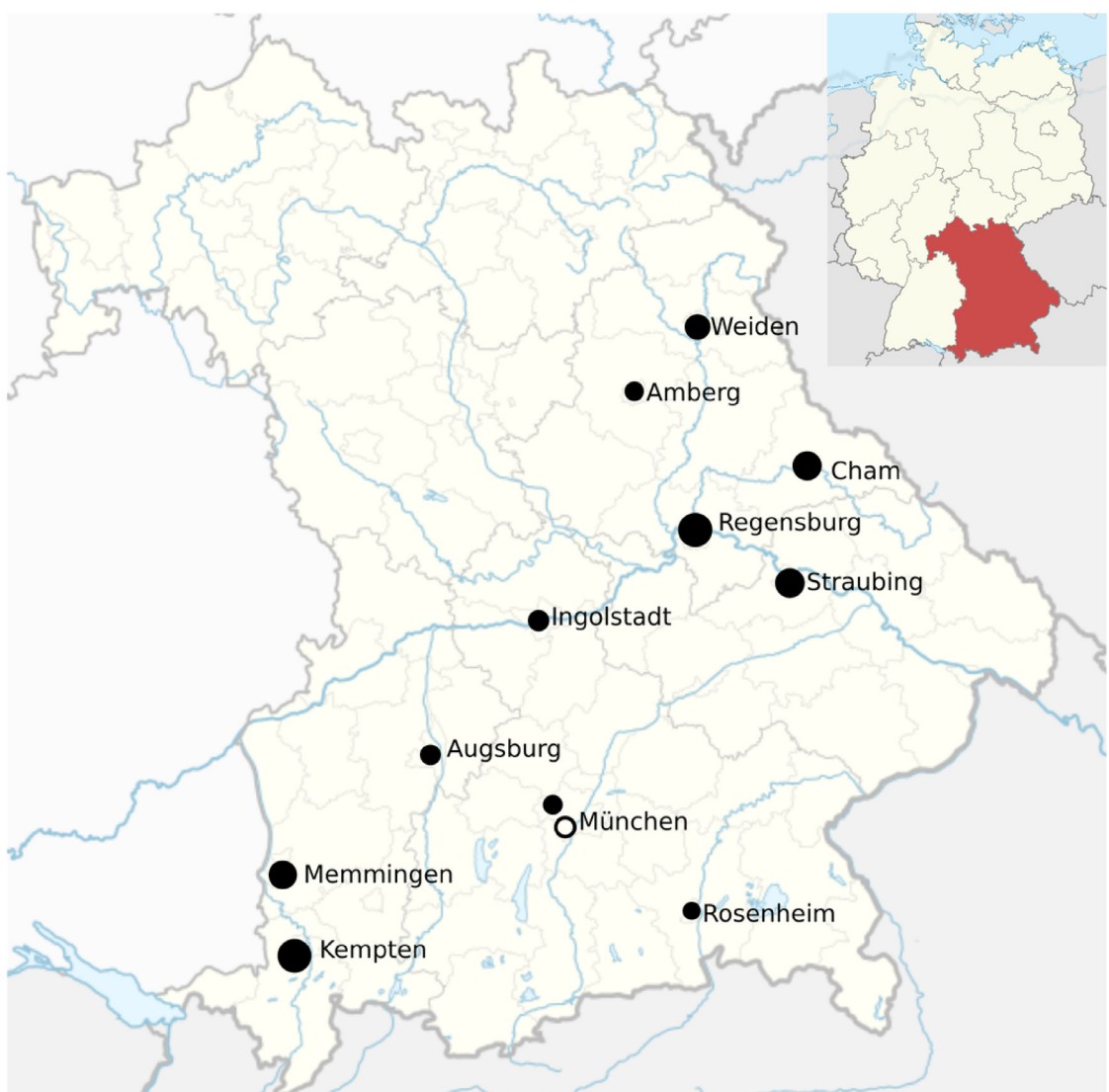

**Fig 1. Locations of participating EDs.** The map locates the participating EDs within Bavaria, South-Eastern Germany. EDs participating in both seasons are depicted as filled dots, unfilled dots represent EDs participating in 2020/2021 only. Dot size is proportional to the number of returned surveys from each site.

2021 were more likely to agree that influenza poses a heavy burden on ED resources. With regard to ethical questions related to influenza and patient care, we observed significant differences between unvaccinated and vaccinated respondents (2020/2021), whereas answers between 2016/2017 and 2020/2021 respondents were virtually the same. In 2020/2021, vaccinated respondents were significantly more likely to agree that influenza vaccination should be mandatory and that no-shows during an influenza pandemic should be laid off. In contrast, unvaccinated respondents were significantly more likely to agree (i) that a mandatory vaccination policy would be a reason to change jobs, (ii) that it would be ethical not to go to work during an influenza pandemic and (iii) that work attendance should be left to the employee. Of note, both groups agreed to a high percentage on a moral obligation to care for influenza infected patients (Fig 4).

**Table 1. Factors associated with influenza vaccination status and socio-demographics of participants.**

| Characteristics | univariate both years | | | | | |
|---|---|---|---|---|---|---|
| | 2016/2017 N = 142[1] | 2020/2021 N = 190[1] | p-value[2] | | | |
| vaccinated this season | **44/142 (31%)** | **112/190 (59%)** | **<0.001** | | | |
| vaccination available on site (ED) | **27/141 (19%)** | **94/189 (50%)** | **<0.001** | | | |
| ≥ 2 times vaccinated | **56/142 (39%)** | **110/190 (58%)** | **0.001** | | | |
| gender | | | 0.5 | | | |
| male | 57/136 (42%) | 68/183 (37%) | | | | |
| female | 79/136 (58%) | 115/183 (63%) | | | | |
| age | | | | | | |
| >44 | 45/38 (33%) | 58/190 (31%) | >0.9 | | | |
| profession | | | 0.069 | | | |
| nurses | 75/142 (53%) | 102/190 (54%) | | | | |
| other | 19/142 (13%) | 12/190 (6%) | | | | |
| physician/student | 48/142 (34%) | 76/190 (40%) | | | | |
| **Characteristics** | **univariate 2020/2021** | | | **multivariate 2020/2021** | | |
| | not vaccinated N = 78[1] | vaccinated N = 112[1] | p-value[2] | OR[3] | 95% CI[4] | p-value[2] |
| vaccination available on site (ED) | **31/77 (40%)** | **63/112 (56%)** | **0.044** | **3.96** | **1.42, 12.2** | **0.011** |
| ≥ 2 times vaccinated | **17/78 (22%)** | **93/112 (83%)** | **<0.001** | **21.3** | **7.41, 73.4** | **<0.001** |
| intending to be vaccinated because of COVID-19 | **2/75 (3%)** | **55/111 (50%)** | **<0.001** | **86.7** | **18.8, 688** | **<0.001** |
| profession | | | **<0.001** | | | |
| nurses | **61/78 (78%)** | **41/112 (37%)** | | - | - | |
| other | **3/78 (4%)** | **5/112 (5%)** | | 1.39 | 0.11, 15.2 | 0.8 |
| physician/student | **14/78 (18%)** | **66/112 (59%)** | | **5.39** | **1.83, 17.2** | **0.003** |

[1] Statistics presented: n/N (%)

[2] Statistical test performed: chi-square test of independence

[3] Odds Ratio

[4] CI = Confidence Interval

## Discussion

Health care workers are frequently exposed to contagious agents. In particular, the close and extended contact to vulnerable patients and individuals with acute respiratory infections like influenza in the ED leads to an enhanced risk of transmitting nosocomial diseases [9–11]. Even though vaccination of ED personnel and health care workers in general is considered to be a reliable and effective method to decrease nosocomial influenza infections and is recommended by the German standing vaccination committee (STIKO) [22, 23] the vaccination coverage within hospitals can still be improved.

In order to analyze the acceptance of influenza immunization, we conducted a survey to explore influenza vaccination coverage and attitudes towards influenza and influenza vaccination among emergency department personnel and found that self-reported vaccination coverage significantly increased from 31% in the 2016/2017 season to 59% in the 2020/2021 season, i.e. before and during the COVID-19 pandemic. These numbers may seem low but are in the range reported previously by similar studies where self-reported seasonal influenza vaccination coverage among health care workers, ED staff and emergency medical services (EMS) personnel was around 40% (28%–54%) [4, 9, 11, 24–26]. Along with the higher vaccination rate in 2020/2021 compared to 2016/2017 we observed a significantly increased proportion of

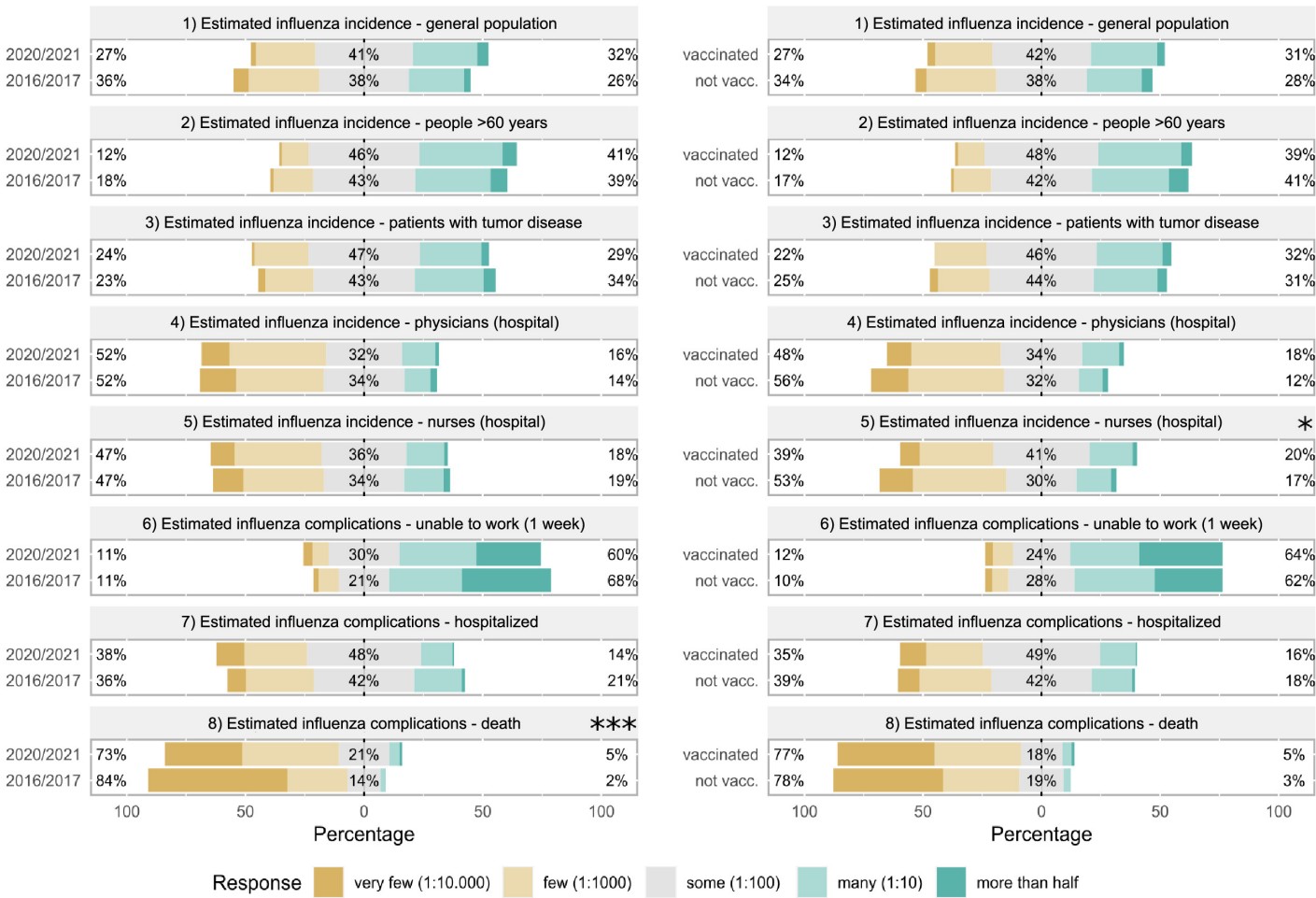

**Fig 2. Comparison of estimated frequencies of influenza infections and complications between 2016/2017 and 2020/2021 or between vaccinated and unvaccinated respondents in 2020/2021.** Estimated frequency of influenza infection and its complications rates were assessed on a 5-Point Likert scale and compared between the seasons 2016/2017 and 2020/2021 (left) or between vaccinated and not vaccinated respondents in 2020/2021 (right). Statistics were performed after conversion of ordinal answer levels to a numeric scale using Mann-Whitney U tests, p-values are depicted as *<0.05, **<0.01 and ***<0.001.

employees reporting availability of influenza vaccination on site (in the ED) and a history of at least two influenza vaccinations. Beyond that, we found no major changes in the data over time, except for the fact that employees in 2020/2021 we more likely to agree that influenza poses a serious threat for ED patients and considered fatal complications of influenza infections to occur more frequently. It is tempting to speculate that the close proximity to the pandemic caused by another respiratory virus might have influenced the assessment. Notably, during the same period, the perception of vaccination associated side effects, as well as attitudes towards more ethical questions related to influenza and patient care remained unchanged. Based on these observations we studied factors linked to the vaccination status in the 2020/2021 season in an univariate and multivariate analysis. We hypothesized that the COVID-19 pandemic (among other reasons) would positively affect the influenza vaccination rate among ED personnel. Indeed, answers that were positively associated with being vaccinated were the availability of influenza vaccination on site (in the ED), a history of at least two influenza vaccinations, profession (being a physician or medical student) and the intention to get vaccinated because of COVID-19. Higher vaccination coverage rates among health care

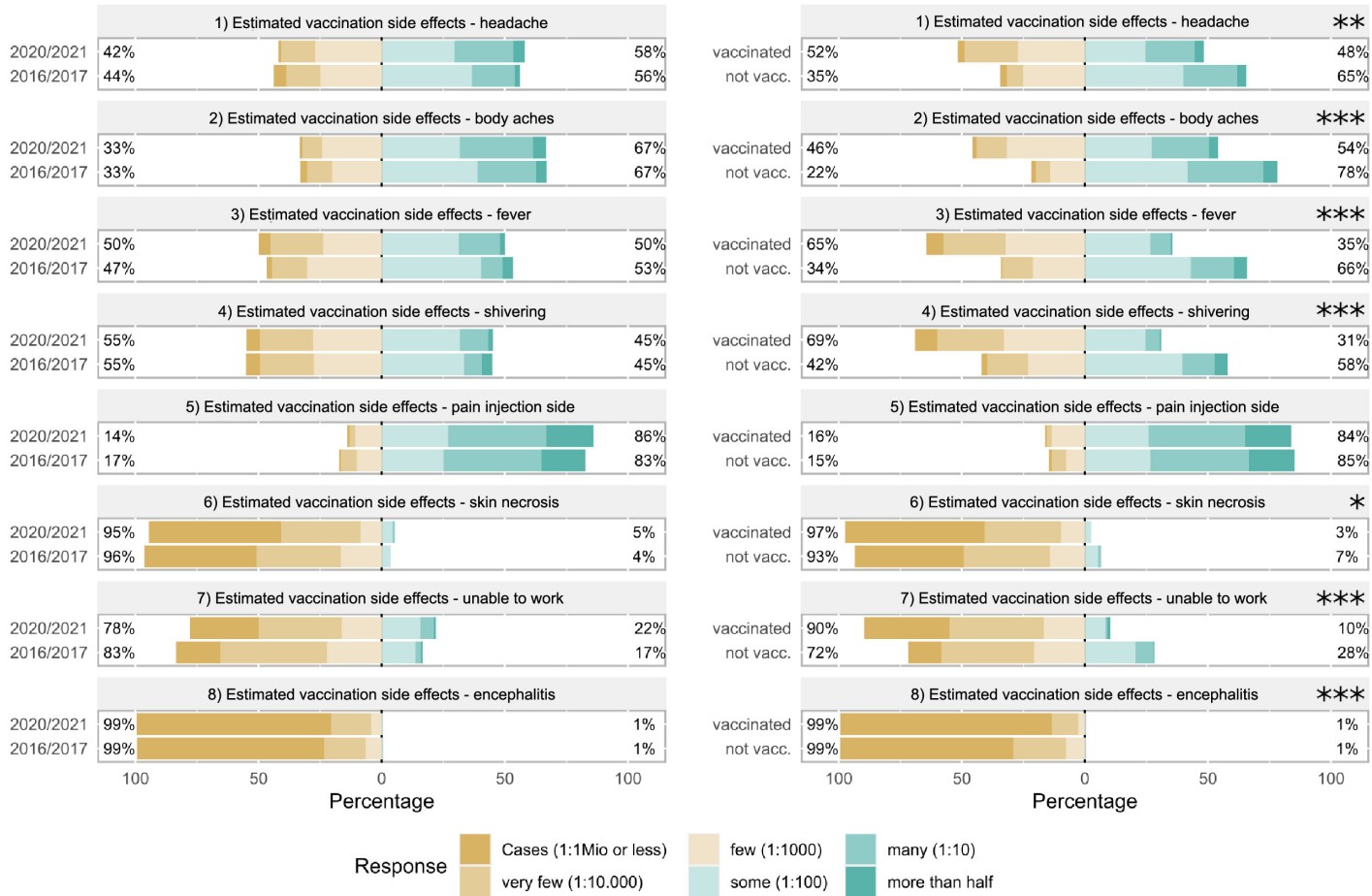

**Fig 3. Comparison of estimated frequencies of influenza vaccination side effects between 2016/2017 and 2020/2021 or between vaccinated and unvaccinated respondents in 2020/2021.** Estimated frequency of side effects associated with influenza vaccination assessed on a 6-Point Likert scale was compared between the seasons 2016/2017 and 2020/2021 (left) or between vaccinated and not vaccinated respondents in 2020/2021 (right). Statistics were performed after conversion of ordinal answer levels to a numeric scale using Mann-Whitney U tests, p-values are depicted as *<0.05, **<0.01 and ***<0.001.

workers due to on site vaccination services were also demonstrated in previous studies with an increase in vaccination coverage rates between +4% and +29% [27–31]. Profession and vaccination history have been associated with being vaccinated against influenza before [26, 32]. A monocentric study from Switzerland reported odds ratios of 7.7 and 9.5, respectively [25]. During the COVID-19 pandemic, Wang et al. also observed a more positive attitude towards influenza vaccination among nurses possibly based on a heightened awareness of the risks associated with airborne infections [16]. In the course of the current increased awareness, targeted education and information about the mode of action and safety of the influenza vaccination is of particular importance to eliminate misleading media reports and doubts about the effectiveness of vaccination. In addition, promotional activities and campaigns in newsletters, salary supplements, mailings, posters, or fact sheets support achieving higher vaccination coverage rates, as reported in other studies with increases ranging from +12% to +26% [33–35]. Another very effective tool to increase vaccination rates among health care workers is a change in policy with mandatory vaccination [31, 36–39]. In studies conducted in the United States or Australia, increases in vaccination coverage rates ranged from +24% to +46%. Although the results are impressive, mandatory influenza vaccination is not an option in many health care

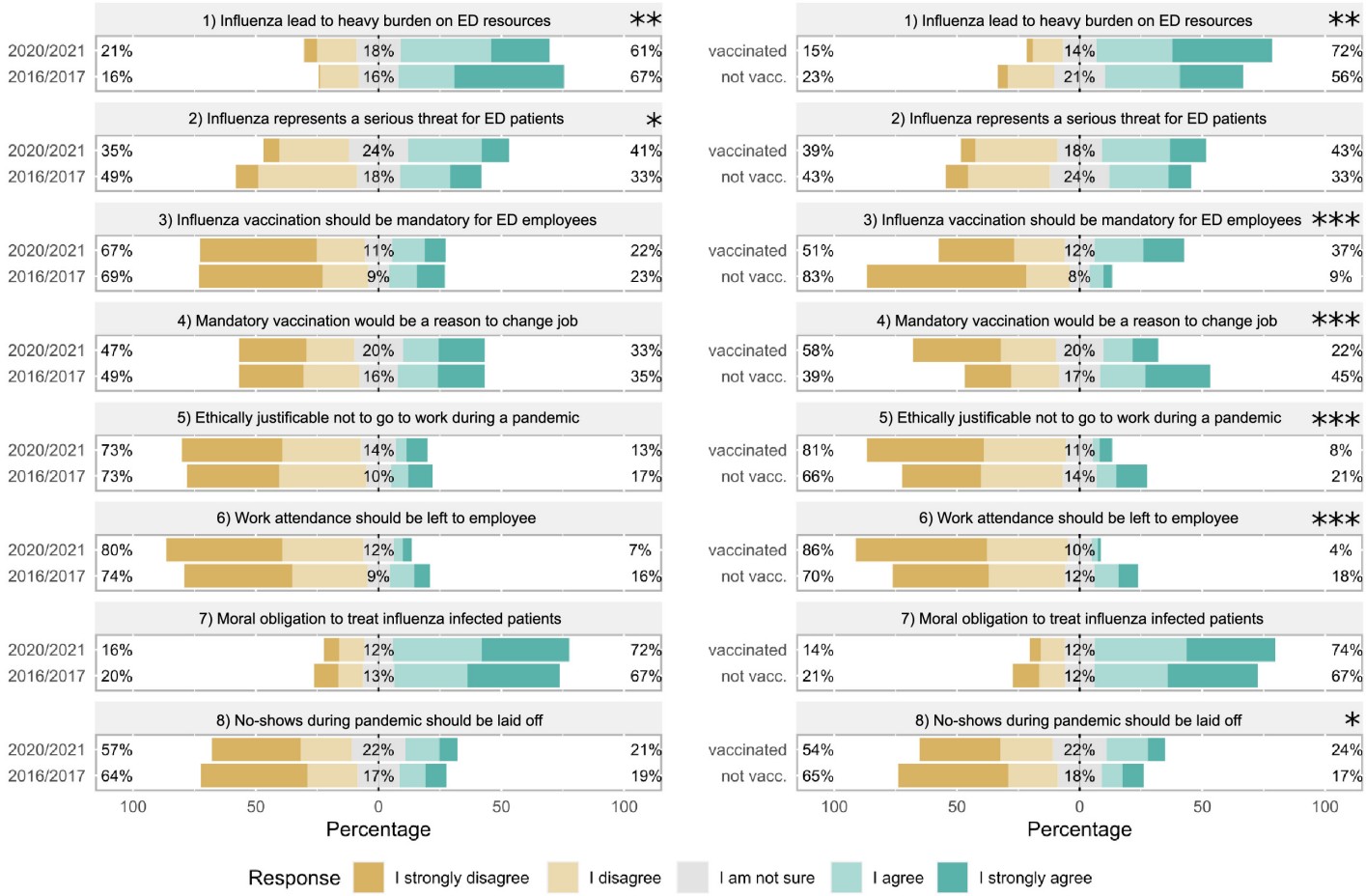

**Fig 4. Comparison of views on the burden of influenza in the EDs and ethical attitude of employees towards vaccination and work attendance.** Acceptance of statements on the impact of influenza on EDs and various ethical issues related to influenza (vaccination) or work attendance assessed on a 5-Point Likert scale was compared between the seasons 2016/2017 and 2020/2021 (left) or between vaccinated and not vaccinated respondents in 2020/2021 (right). Statistics were performed after conversion of ordinal answer levels to a numeric scale using Mann-Whitney U tests, p-values are depicted as *<0.05, **<0.01 and ***<0.001.

systems including Germany (influenza vaccination is free but not mandatory for hospital employees).

With regard to questions pertaining to influenza incidence and complications, we found no major differences between vaccinated and not vaccinated respondents in 2020/2021, apart from higher estimates for risk of infection for nurses among vaccinated employees compared to unvaccinated respondents. In addition, unvaccinated ED personnel estimated significantly higher frequencies of vaccination side effects compared to their vaccinated colleagues. Together with concerns about vaccine effectiveness [11, 16, 17, 25, 26, 40], concerns about side effects and vaccine safety have been identified as a major obstacle for vaccination of health care workers before [24–26, 40]. Similarly, belief in vaccine safety and effectiveness was higher in vaccinated than in unvaccinated EMS personnel [11]. Another concern previously raised by ED staff was that illness could be caused by the vaccine [9].

This study also examined the association of ethical attitudes and influenza vaccination, which provides an important and unique contribution to influenza vaccination research. In our study, vaccinated and unvaccinated ED personnel in 2020/2021 also differed in their attitudes towards ethical questions raised. Whereas both groups agreed to a high percentage on a

moral obligation to care for influenza patients and assessed influenza as a heavy burden on their resources, unvaccinated respondents were less likely to accept lay-offs for employees not showing up for work and rather stated that work attendance should be the employee's decision (both during a potential influenza pandemic). Vaccinated participants on the other hand, would rather agree that vaccination should be mandatory and were less likely to consider job changes in the event of a mandatory vaccination policy. A significant role for employers' vaccination policy but low support for mandatory vaccination has also been reported from emergency medical personnel before [11].

Several aspects might limit the generalizability of our findings. The study is not representative for all EDs in the surveyed area of Bavaria, Germany, and only a part of the employees answered the questionnaire. The response rate of 38% in the 2020/2021 season may seem low but can be explained as of the higher number of questionnaires sent out to the emergency departments than in the 2016/2017 season. In general, the number of returned questionnaires in the 2020/2021 season is higher than the number in the 2016/2017 season. With regard to the distribution of the questionnaire, it must be noted that the respective ED leaders were responsible for the process and therefore multiple responses from a single employee cannot be ruled out.

## Conclusion

Our results suggest that not only the assessment of influenza as a more serious risk during the current COVID-19 pandemic but also ease of access to vaccination is crucial for an increased influenza vaccination rate. In order to increase the influenza vaccination coverage among ED personnel, a broad effort to offer influenza vaccination on site in the ED might help. Given the observed differences in the perception of vaccine side effects among vaccinated and not vaccinated personnel, interventions targeting not only influenza disease but specific knowledge about vaccination and its side effects could help to overcome prejudices.

## Acknowledgments

We would like to thank and acknowledge the work and crucial support by Christine Fuhrmann, Universitätsklinikum Regensburg; Dr. Rupert Grashey, Klinikum Memmingen; Dr. Andrea Dauber, Klinikum Weiden; Dr. Andreas Pohl, Klinikum Weiden; Dr. Florian Demetz, München Klinik Harlaching; Dr. Michael Bayeff-Filloff, Klinikum Rosenheim; PD Dr. Markus Wehler, Klinikum Augsburg; Dr. Marc Bigalke, Klinikum St. Marien Amberg; Dr. Kathrin Tzaferidis, Klinikum Dritter Orden München; Dr. Dagmar Strauß, Klinikum Kempten; Dr. Christian Thiel, Klinikum St. Elisabeth Straubing; Dr. Thomas Etti, Krankenhaus Cham and Dr. Stephan Steger, Klinikum Ingolstadt.

## Author Contributions

**Conceptualization:** Anna-Maria Stöckeler, Markus Zimmermann, Frank Hanses.

**Data curation:** Anna-Maria Stöckeler.

**Formal analysis:** Frank Hanses.

**Methodology:** Anna-Maria Stöckeler, Frank Hanses.

**Project administration:** Frank Hanses.

**Resources:** Markus Zimmermann.

**Supervision:** Markus Zimmermann, Frank Hanses.

**Visualization:** Anna-Maria Stöckeler, Philipp Schuster, Frank Hanses.

**Writing – original draft:** Anna-Maria Stöckeler, Philipp Schuster, Frank Hanses.

**Writing – review & editing:** Anna-Maria Stöckeler, Philipp Schuster, Markus Zimmermann, Frank Hanses.

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
