## [Decision Letter · Decision Letter 0]

20 Jul 2021

PONE-D-21-08907

Influenza vaccination coverage among emergency department personnel is associated with perception of vaccination and side effects, vaccination availability on site and the COVID-19 pandemic

PLOS ONE

Dear Dr. Hanses,

Thank you for submitting your manuscript to PLOS ONE. After careful consideration, we feel that it has merit but does not fully meet PLOS ONE’s publication criteria as it currently stands. Therefore, we invite you to submit a revised version of the manuscript that addresses the points raised during the review process.

As per the reviewers' reports, a number of methodological clarifications are required in order for your submission to meet our third publication criterion (https://journals.plos.org/plosone/s/criteria-for-publication#loc-3). Please ensure that you carefully respond to each of the points the reviewers have raised when preparing your revisions.

We look forward to receiving your revised manuscript.

Kind regards,

Jamie Males

Staff Editor

PLOS ONE

Journal Requirements:

2.  Please provide additional details regarding participant consent. In the ethics statement in the Methods and online submission information, please ensure that you have specified:

 - whether consent was obtained

 - whether consent was informed

 - what type of consent you obtained (for instance, written or verbal, and if verbal, how it was documented and witnessed).

 - if your study included minors, state whether you obtained consent from parents or guardians.

 - if the need for consent was waived by the ethics committee, please include this information.”

3. Please provide the names of the participating EDs.

4. please provide the date ranges when surveys were collected during the two seasons.

Reviewers' comments:

Reviewer's Responses to Questions

**Comments to the Author**

1. Is the manuscript technically sound, and do the data support the conclusions?

Reviewer #1: Partly

Reviewer #2: Yes

2. Has the statistical analysis been performed appropriately and rigorously? 

Reviewer #1: Yes

Reviewer #2: Yes

3. Have the authors made all data underlying the findings in their manuscript fully available?

Reviewer #1: No

Reviewer #2: No

4. Is the manuscript presented in an intelligible fashion and written in standard English?

Reviewer #1: Yes

Reviewer #2: Yes

5. Review Comments to the Author

Reviewer #1: The authors present the results of two cross-sectional surveys of influenza vaccination uptake among ED personnel in Bavaria, Germany. The manuscript is well-written for the most part and the study findings are generally of interest; the methods require some clarification and the discussion section needs to be reworked. Specific suggests are as follows:

Abstract, line 27: The extremely low response rate in 2020-2021 is a threat to the validity of your findings and should be noted in the limitations section (which is missing from the paper).

Abstract, line 33: The pandemic is not a factor that differs across the study population; this could be tweaked to say something like "the effect of the pandemic on influenza vaccination intentions" or similar.

Introduction, line 52: Usually influenza vaccination is also recommended for those at increased risk of complications from influenza disease, regardless of their risk of acquiring influenza -- this could be noted here as well.

Methods, line 76: Given that the article presents a comparison with findings in the COVID-19 pandemic context and the earlier findings, it would be useful to include a brief note here or in the introduction about the impetus for the initial 2016-2017 survey -- i.e. was it motivated by a specific data point, a policy change, some other global context?

Methods, line 77: Are these 14 EDs the universe of EDs in Bavaria? If not, how many are there/what percentage is represented by the 14 invited, and how were they selected for recruitment?

Methods, lines 89-99: The sample selection requires substantial clarification. Were the 229 and 498 staff surveyed in each season all of the staff working in the ED -- or all of the physicians, nurses, and 'administrative' staff? (And if the latter, which types of staff does this exclude?) If this was not 100% of the staff in the targeted EDs, what proportion is represented by the disseminated and returned surveys? Also, please provide more detail on how specifically the surveys were disseminated -- e.g. were they mailed to individual staff? Was a package of surveys sent to the ED and distributed by a chief/leader? Was the package sent and the surveys were left on a table in the lunch room? Etc. If surveys were not addressed to individuals and they were returned anonymously, how did the authors assure that only targeted staff (from the ED) completed and returned surveys and that multiple surveys were not completed by a single person?

Results, Table 1: Please change "vaccinated because of COVID-19" to "intending to be vaccinated because..." or similar. As written it is confusing to see entries in the 'unvaccinated' category for this variable as the unvaccinated people clearly were not vaccinated because of COVID-19. I will also note that I found this table extremely difficult to follow; you might consider reformatting it or even breaking it up into multiple tables if the space allotted by the journal permits.

Results, line 148: The fact that the comparisons of unvaccinated with vaccinated persons include only data from 2020-21 should be noted at every point in the manuscript or tables/figures where these comparisons are made.

Results, line 157: Higher risk relative to what, or whom? E.g. higher than the general public, higher than doctors...?

Results, line 160: For this and all of the figures, the figure titles need to be more descriptive so that they can be interpreted without reading the text. I would also suggest including in the figure title that the figures compare 2016-17 to 2020-21 and compare unvaccinated to vaccinated persons.

Results, line 169: How was this list of possible vaccine side effects selected? Skin necrosis and encephalitis are essentially nonexistent as a result of influenza vaccination, whereas mild systemic symptoms and injection site pain would commonly be expected. Also, how were these presented in the survey? Listing very serious and extremely rare side effects alongside mild and common side effects could potentially bias respondents' selections.

Results, line 184: As noted above, you would need to emphasize here that this comparison was only done for 2020-2021 -- however, did you attempt to combine data across seasons to see if the same associations with vaccination status persist? (I assume this comparison was not made for 2016-2017 due to the small sample of people vaccinated; however, this could be clarified in the text as well.)

Discussion, general comment: The discussion substantially reiterates the introduction and results sections. Repetitive content should be streamlined so additional interpretation of findings can be excluded. Areas that seem redundant and could be deleted entirely or reduced to a single sentence include lines 213-217 and 223-230.

Discussion, lines 201-206: This should be noted in detail in the introduction as context and then briefly restated here in the discussion if needed.

Discussion, lines 207-212: Some of this is already in the introduction and could be deleted; the parts that are not in the introduction already might more properly belong there as they provide justification for the importance of this study.

Discussion, lines 231-234: It would be helpful here to include some suggested actions to take based on these findings or at least the implications of the findings as nurses presumably provide substantial amounts of care in the ED and are at least equally as much at risk of acquiring and transmitting influenza as physicians.

Discussion, lines 236-239: It's confusing when you bounce back and forth between the comparison across time and the comparison of unvaccinated with vaccinated personnel in the 2020-2021 season. I would recommend a discussion paragraph talking about differences and similarities over time, and then a separate paragraph discussing differences observed by vaccination status in 2020-2021.

Discussion, lines 249-250: Beyond the question of vaccination mandates, I have never seen a published study that examines ethical attitudes and their association with influenza vaccination before -- at least not with a suite of items used together like this. In my view, this is an unique and significant contribution of your study. I would create a separate paragraph focusing on the ethics questions/findings and implications for pandemic preparedness, vaccination promotion, etc.

Discussion, line 259: As noted above, the manuscript lacks a limitations section; one should be added here.

Conclusion, line 261: I did not see a measure of awareness in this study; this should be rephrased to reflect what was actually asked in the survey.

Conclusion, lines 263-264: This finding is extremely important for increasing rates of influenza vaccination in Bavarian EDs because it is actionable and the necessary intervention is obvious. I strongly suggest adding to your discussion a supporting paragraph that discusses the change in onsite vaccine offering over time, any national or regional policies or practices that might support or be a barrier to this (e.g. do Germans have nationalized or private healthcare that might complicate payment; is there a national or regional policy to provide free influenza vaccine to healthcare workers; is there a national or regional target for influenza vaccination, etc.). It might be useful to cite the findings of the U.S. Task Force on Community Preventive Services, which found strong evidence that offering vaccination onsite increases vaccination coverage. Although the Task Force is convened in the U.S., the evidence considered is global and the findings are certainly applicable to the Bavarian context: https://www.thecommunityguide.org/findings/worksite-health-seasonal-influenza-vaccinations-healthcare-on-site. In conjunction with this, some of the studies in other countries noting the association of onsite offering of vaccine with vaccination uptake among HCP could be included.

Conclusion, lines 266-267: I am not aware of any evidence that education/communication campaigns when used alone are effective in increasing vaccination uptake, so I would not term these "most beneficial". I suggest recommending this type of effort in conjunction with onsite offering of influenza vaccine as discussed above.

Figures 2-4: The labels on individual figure entries need to be edited to be more specific as many are not interpretable without a copy of the survey instrument - and this would not be helpful to readers who do not speak German. I would recommend an appendix with the translated survey instrument, but if this is cost- or time-prohibitive, the labels for each entry in the figure should be written as complete phrases or sentences that capture what was actually presented to survey respondents, e.g. "I believe influenza results in death at this frequency" instead of "influenza complications - death".

Reviewer #2: This article is well written. Methodology seems sound. Conclusions were pretty predictable.

I have three questions :

1- Could the authors please elaborate on the fact that vaccination was not available on sit everywhere ? It seems odd that it is not the case in every setting. Could you please describe what health care workers have to do to be vaccinated when flu vaccine is not available on site ?

2- Do they authors have the 2020/2021 national vaccination coverage in physicians and nurses as a comparator ?

3- Could you precise others categories than administrative personnel in the non physicians or nurses people ? Did you survey other personnels in contact with patients such as cargivers known for their low vaccine coverage (even lower than nurses’ one) ?

6. PLOS authors have the option to publish the peer review history of their article (what does this mean?). If published, this will include your full peer review and any attached files.

Reviewer #1: No

Reviewer #2: **Yes: **Paul Loubet

---

## [Author Response · Author response to Decision Letter 0]

2 Sep 2021

Journal Requirements:

1. Please ensure that your manuscript meets PLOS ONE's style requirements, including those for file naming. …

We have checked PLOS ONE´s style requirements again and hope everything is correct now.

2. Please provide additional details regarding participant consent …

We thank you for pointing this out. As the data were collected anonymously, self-administered and on a voluntary basis, no consent was obtained for this study. For more details please see lines 110-112.

3. Please provide the names of the participating EDs.

Thank you for this suggestion. We have made the change and added the names of the participating EDs. Please see lines 95-98.

4. please provide the date ranges when surveys were collected during the two seasons.

We apologize for not pointing this out. We have corrected the methods part and you can find the date ranges in lines 86-87 as well as in lines 90-91.

5. In your Data Availability statement, you have not specified where the minimal data set underlying the results described in your manuscript can be found. …

This observation is correct. An anonymized version of the dataset is available from zenodo.org

(https://doi.org/10.5281/zenodo.5164388). "

Reviewers' comments:

Reviewer 1:

Abstract, line 27: The extremely low response rate in 2020-2021 is a threat to the validity of your findings and should be noted in the limitations section (which is missing from the paper).

We thank the reviewer for pointing this out. We have added a limitation section at the end of the discussion, which, among others, concerns to the response rate in 2020/2021. Please see lines 339-342.

Abstract, line 33: The pandemic is not a factor that differs across the study population; this could be tweaked to say something like "the effect of the pandemic on influenza vaccination intentions" or similar.

We agree and have updated this sentence. Please see line 31.

Introduction, line 52: Usually influenza vaccination is also recommended for those at increased risk of complications from influenza disease…

Thank you for this important observation. We have changed this section. Please see lines 54 and 60-61.

Methods, line 76: Given that the article presents a comparison with findings in the COVID-19 pandemic context and the earlier findings, it would be useful to include a brief note … about the impetus for the initial 2016-2017 survey…

We have revised the text to address your concerns and explained in more detail the impetus for the first survey in 2016/2017 and the linkage to the second survey. Please see lines 85-93.

Methods, line 77: Are these 14 EDs the universe of EDs in Bavaria? If not, how many are there/what percentage is represented by the 14 invited, and how were they selected for recruitment?

We thank the reviewer for pointing this out. We added this information to the limitations at lines 337-339.

Methods, lines 89-99: The sample selection requires substantial clarification. … Also, please provide more detail on how specifically the surveys were disseminated ...

We agree and have corrected the sample recruitment part with a more detailed explanation of the process. Please see lines 113-117. We also revised the text to include your concerns about multiple surveys in the new limitations section. Please see lines 343-345.

Results, Table 1: Please change "vaccinated because of COVID-19" to "intending to be vaccinated because..." … you might consider reformatting it or even breaking it up into multiple tables …

We have made the changes and thank the reviewer for these suggestions. Please see lines 165-171.

Results, line 148: The fact that the comparisons of unvaccinated with vaccinated persons include only data from 2020-21 should be noted at every point in the manuscript or tables/figures where these comparisons are made.

We have taken note of this important observation and added the note in the certain text passages. Please see lines 29,36,199,204, 215,220,223,224,239,327.

Results, line 157: Higher risk relative to what, or whom

We agree that this information is missing and added the comparison group in lines 190-191.

Results, line 160: …the figure titles need to be more descriptive so that they can be interpreted without reading the text. I would also suggest including in the figure title that the figures compare 2016-17 to 2020-21 and compare unvaccinated to vaccinated persons.

We´ve revised the figure titles to become more descriptive and changed the figure titles as suggested. Please see lines 194-201, 210-217, 233-240.

Results, line 169: How was this list of possible vaccine side effects selected… Also, how were these presented in the survey? Listing very serious and extremely rare side effects alongside mild and common side effects could potentially bias respondents' selections.

We thank the reviewer for pointing this out. The list of side effects was loosely built on previous studies, which we added in the method part (query and variables) in lines 129-130. The order of the side effects listing in the paper is almost the same than in the questionnaires. Also, we thank you for your concern regarding the alongside listing of mild, extreme and almost nonexistent side effects listings. We are aware that some effects are very rare. However, this was done intentionally to determine whether effects were partially overestimated.

Results, line 184: … did you attempt to combine data across seasons to see if the same associations with vaccination status persist? …

This observation is correct and the comparison would be interesting, but in 2016/2017 season, the items pertaining to the SARS-CoV-2 pandemic was not included so this variable would be missing. We explained that in more detail in lines 148-150.

Discussion, general comment: … Areas that seem redundant and could be deleted entirely or reduced to a single sentence include lines 213-217 and 223-230.

We thank the reviewer for this suggestion. We agree and have changed this part of the discussion. Please see lines 263-268 and 282-285.

Discussion, lines 201-206: This should be noted in detail in the introduction as context and then briefly restated here in the discussion if needed.

We´ve corrected this part of the discussion as we agree on the reviewers suggestion. Please see lines 54-60, 64-67 and 249-254.

Discussion, lines 207-212: Some of this is already in the introduction and could be deleted…

We revised the discussion and introduction part and thank the author for this suggestion. Please see lines 51-52, 243-249 and 255-259.

Discussion, lines 231-234: It would be helpful here to include some suggested actions to take based on these findings …

We thank the reviewer for this comment and have added a section with suggested actions for a higher vaccination rate. Please see lines 297-308.

Discussion, lines 236-239: … I would recommend a discussion paragraph talking about differences and similarities over time, and then a separate paragraph discussing differences observed by vaccination status in 2020-2021.

We agree that this part of the discussion was confusing. We have revised the structure of the discussion as the reviewer suggested. Please see lines 271-290 and 310-317.

Discussion, lines 249-250: I would create a separate paragraph focusing on the ethics questions/findings and implications for pandemic preparedness, vaccination promotion, etc.

We thank the reviewer for this pleasing assessment. We have extended the ethical findings sections. Please see lines 325-330.

Discussion, line 259: As noted above, the manuscript lacks a limitations section; one should be added here.

We apologize for the missing limitation section, which was now included in the last part of the discussion. Please see lines 337-345

Conclusion, line 261: I did not see a measure of awareness in this study; this should be rephrased to reflect what was actually asked in the survey.

We have corrected the term and changed it into the assessment of influenza as a more serious risk. Please see lines 348-349.

Conclusion, lines 263-264: … I strongly suggest adding to your discussion a supporting paragraph that discusses the change in onsite vaccine offering over time…

We thank the reviewer for his important notes and added some reference papers to our study. Please see lines 290-293. 

Conclusion, lines 266-267: … I would not term these "most beneficial". I suggest recommending this type of effort in conjunction with onsite offering of influenza vaccine as discussed above.

We have changed the text according to the reviewer´s suggestion. Please see line354-355

Figures 2-4: The labels on individual figure entries need to be edited to be more specific as many are not interpretable without a copy of the survey instrument …

We agree to the reviewer´s comment and described the items on the figures in more detail. Please see Figures 2-4.

Reviewer 2: 

I have three questions :

1- Could the authors please elaborate on the fact that vaccination was not available on sit everywhere? … 

Thank you for your questions, which we would like to address in more detail. In Germany, vaccinations are generally free of charge for hospital employees. However, vaccinations do not take place exclusively in the ED, but can also be given, for example, via an appointment with the company doctor.

2- Do they authors have the 2020/2021 national vaccination coverage in physicians and nurses as a comparator ? 

Unfortunately, the requested data is not yet available.

3- Could you precise others categories than administrative personnel in the non physicians or nurses people ? Did you survey other personnels in contact with patients such as cargivers known for their low vaccine coverage?

We thank the reviewer for this important question. The others category only includes non medical staff (mainly administrative). Caregivers were not included in this survey, only registered nurses.

---

## [Decision Letter · Decision Letter 1]

5 Nov 2021

Influenza vaccination coverage among emergency department personnel is associated with perception of vaccination and side effects, vaccination availability on site and the COVID-19 pandemic

PONE-D-21-08907R1

Dear Dr. Hanses,

We’re pleased to inform you that your manuscript has been judged scientifically suitable for publication and will be formally accepted for publication once it meets all outstanding technical requirements. As one of the previous reviewers (Reviewer #1), I found your revision to be extremely responsive to all of my comments and I appreciate your time in carefully addressing each of the stated concerns. The revised manuscript provides the necessary context for your findings and is organized in a way to highlight the comparisons made over time as well as by vaccination status and the unique contribution of the ethical/values data gathered. 

Kind regards,

Megan C. Lindley

Guest Editor

PLOS ONE

Additional Editor Comments (optional):

The authors' conscientious response to the previous review comments is much appreciated. The revised manuscript is much improved and highlights the unique contributions of the authors' research.

Reviewers' comments:

Reviewer's Responses to Questions

**Comments to the Author**

1. If the authors have adequately addressed your comments raised in a previous round of review and you feel that this manuscript is now acceptable for publication, you may indicate that here to bypass the “Comments to the Author” section, enter your conflict of interest statement in the “Confidential to Editor” section, and submit your "Accept" recommendation.

Reviewer #2: All comments have been addressed

2. Is the manuscript technically sound, and do the data support the conclusions?

Reviewer #2: Yes

3. Has the statistical analysis been performed appropriately and rigorously? 

Reviewer #2: Yes

4. Have the authors made all data underlying the findings in their manuscript fully available?

Reviewer #2: Yes

5. Is the manuscript presented in an intelligible fashion and written in standard English?

Reviewer #2: Yes

6. Review Comments to the Author

Reviewer #2: Thank you for having adressed my comments. I have no other comments

................................

7. PLOS authors have the option to publish the peer review history of their article (what does this mean?). If published, this will include your full peer review and any attached files.

Reviewer #2: **Yes: **Paul Loubet